# Polycyclic Aromatic Hydrocarbons Detected in Processed Meats Cause Genetic Changes in Colorectal Cancers

**DOI:** 10.3390/ijms222010959

**Published:** 2021-10-11

**Authors:** Tracie Cheng, Stephanie Chaousis, Sujani Madhurika Kodagoda Gamage, Alfred King-yin Lam, Vinod Gopalan

**Affiliations:** 1Cancer Molecular Pathology, School of Medicine & Dentistry, Griffith University, Gold Coast 4222, Australia; tracie.cheng@griffith.edu.au (T.C.); schaousis@gmail.com (S.C.); s.kodagodagamage@griffith.edu.au (S.M.K.G.); 2Department of Anatomy, Faculty of Medicine, University of Peradeniya, Kandy 20404, Sri Lanka

**Keywords:** polycyclic aromatic hydrocarbons, colorectal cancer, processed meat, cancer genes

## Abstract

Polycyclic aromatic hydrocarbons (PAHs) are commonly ingested via meat and are produced from high-temperature cooking of meat. Some of these PAHs have potential roles in carcinogenesis of colorectal cancer (CRC). We aimed to investigate PAH concentrations in eight types of commonly consumed ready-to-eat meat samples and their potential effects on gene expressions related to CRC. Extraction and clean-up of meat samples were performed using QuEChERS method, and PAHs were detected using GC-MS. Nine different PAHs were found in meat samples. Interestingly, roast turkey contained the highest total PAH content, followed by salami meat. Hams of varying levels of smokedness showed a proportional increase of phenanthrene (PHEN), anthracene (ANTH), and fluorene (FLU). Triple-smoked ham samples showed significantly higher levels of these PAHs compared to single-smoked ham. These three PAHs plus benzo[a]pyrene (B[a]P), being detected in three meat samples, were chosen as treatments to investigate in vitro gene expression changes in human colon cells. After PAH treatment, total RNA was extracted and rtPCR was performed, investigating gene expression related to CRC. B[a]P decreased mRNA expression of *TP53*. In addition, at high concentrations, B[a]P significantly increased *KRAS* expression. Treatments with 1 µM PHEN, 25 µM, and 10 µM FLU significantly increased *KRAS* mRNA expression in vitro, implying the potential basis for PAH-induced colorectal carcinogenesis. Opposingly, the ANTH treatment led to increased *TP53* and *APC* expression and decreased *KRAS* expression, suggesting an anti-carcinogenic effect. To conclude, PAHs are common in ready-to-eat meat samples and are capable of significantly modifying the expression of key genes related to CRC.

## 1. Introduction

Incidence of colorectal carcinoma (CRC) continues to increase, particularly in countries adopting a Western lifestyle [1]. As CRC is a disease of age and lifestyle, risk factors include diets high in meats and fats and low in fibre and vegetables, higher body-mass indices (BMI), and cigarette smoking [2,3]. Though many studies have been conducted in attempts to understand the mechanistic relationship between meat intake and CRC, there has not been a clear component of meat linked to CRC [4]. The International Agency for Research on Cancer (IARC) pronounced red and processed meat to be Group 1 carcinogens and suggested that polycyclic aromatic hydrocarbons are causative compounds to CRC [5].

Polycyclic aromatic hydrocarbons (PAHs) describe a group of organic compounds containing two or more aromatic rings [6]. They can contaminate meat through direct pyrolysis of food nutrients and from its deposition via smoke produced from incomplete combustion of organic matter [4]; thus, meats that have been smoked or barbequed often have high concentrations of PAHs [7]. Although PAHs are present ubiquitously throughout the environment, the main source of exposure is through the diet [8]. Whilst self-preparation of foods and meats can help to minimise PAH formation [7], processed meats still compose approximately 20% of meat intake of the Australian diet [9]. Once ingested, PAHs are metabolised to form carcinogenic metabolites [10].

Thirteen PAHs investigated in the current study were examined by the IARC and their carcinogenic classifications are detailed in Table 1 [11]. Due to its classification as a group 1 carcinogen, a lot of research has focused on a specific compound, benzo[a]pyrene (B[a]P). B[a]P has been used as a model for PAH metabolism and cancer studies. Briefly, ingested PAHs can ultimately be converted to carcinogenic PAH-diol epoxides via metabolic enzymes [12]. These PAH-diol epoxides can bind DNA and create DNA adducts that interfere with gene transcription and confer genotoxic, mutagenic, and carcinogenic risk [10]. 

The consumption of red and processed meats is linked to colorectal, lung, oesophageal, gastric, and bladder malignancies [13], although it is unclear the exact role of PAHs in these associations. PAHs consumed via ingestion of meat have been primarily linked to cancers of the gastrointestinal tract [14], whilst exposure to environmental PAHs such as tobacco smoking, wood burning, and road traffic have primarily been linked to lung [12] and breast cancers [15]. There are limited studies investigating genetic effects of PAHs in colorectal cancer cells and tissues, however, current available literature investigates the effects of genetic variations i.e., polymorphisms and their risk on CRC [16,17,18]. Current epidemiological data suggests B[a]P (a surrogate PAH) is associated with higher risk of CRC [19,20,21]. Furthermore, the presence of PAH-induced DNA adducts has been shown in the colon [22,23]. Given this evidence for a carcinogenic role of PAHs in CRC, there is limited information regarding PAHs and their potential effects on key genes involved in CRC, such as *TP53*, *APC*, *CTNNB1*, and *KRAS* [24]. Thus, our study aims to use a QuEChERS (quick, easy, cheap, effective, rugged, and safe) approach [25] to extract PAHs from processed meats in Australia to measure their occurrence in these and evaluate their in vitro effects on genes related to colorectal cancer. 

## 2. Results

### 2.1. PAH Detection via GC-MS 

Using the equipment and conditions described in Section 4.4, all 13 PAHs were successfully detected via GC-MS. Appendix A shows the chromatograms produced for each PAH that validates the parameters used in MS/MS. Using five identification points (10, 25, 50, 100, and 200 ppb) per PAH [26], calibrations curves were constructed and detailed in Table 2. As all *r*-values of regression were greater than 0.993, these curve equations were deemed appropriate to use as calibration curves.

### 2.2. Recovery of PAHs Using QuEChERS Extraction

PAHs were spiked at 1000 ng into 6 mL of ACN. The extraction and detection then proceeded accordingly with blank and meat samples. Levels of B[b]F and D[a,h]A, which were detected by GC-MS, are detailed in Table 3. These levels were compared to the unspiked hexane blank, which did not contain detectable levels of these PAHs. B[b]F showed the highest level of recovery at 94% and D[a,h]A was recovered at 74.5%. These recovery values are acceptable and validate the QuEChERS method of PAH extraction in our meat samples. 

### 2.3. PAHs Are Found in Supermarket Meat Samples at Detectable Concentrations

By using QuEChERS method of extraction and clean-up coupled with GC-MS analysis, we detected a variety of PAHs present in common supermarket ready-to-eat meat samples. GC-MS analysis found 9 of 13 PAH analytes present at detectable concentrations in meat samples as detailed in Appendix A. The total concentration of polycyclic aromatic hydrocarbons in these meat samples ranged from 24.04 µg/kg to 90.75 µg/kg. Meat samples in order of increasing PAH concentrations are: single smoked ham, roast chicken, frankfurt, double smoked ham, roast beef, triple smoked ham, salami, and roast turkey (Figure 1). The most abundant PAHs detected across all meat samples were phenanthrene and anthracene. Acenapthylene and chrysene were present only in salami and double-smoked ham, respectively. Fluorene was detected at low levels in most samples except in single-smoked ham and salami, where it was absent and at high levels, respectively. Single-smoked ham contained the lowest total PAH, which was composed only of PHEN and ANTH. Interestingly, B[b]F and B[k]F were highly present in roast turkey. On the other hand, benzo[a]anthracene, indeno[1,2,3-cd]pyrene, dibenzo[a,h]anthracene, and benzo[ghi]perylene were not detected in any meat samples.

### 2.4. Effect of Smoking in Ham on PAH

To investigate the effect of smoking on PAH levels, three types of ready-to-eat ham samples were tested: single-, double-, and triple-smoked hams. As most nitrate-containing cured hams will undergo at least one round of smoking, we could not obtain a non-smoked ham sample for comparison. Noticeably, concentrations of two PAHs, phenanthrene and anthracene, increased as the level of smoking in ham increased. Furthermore, triple-smoked ham contains significantly higher [PHEN] (*p* = 0.035) and [ANTH] (*p* = 0.030) compared to its single-smoked counterpart (Figure 1). This shows a proportional relationship between PHEN and ANTH concentrations in meat with regards to level of smoking. This relationship is noted for fluorene, however, this was not statistically significant. This suggests that increased smoking during ham processing increases the formation of PHEN, ANTH, and to some extent FLU in ham samples.

### 2.5. PAH Induced Genetic Changes In Vitro 

Low-dose B[a]P treatment significantly decreased *TP53* expression in non-neoplastic colon cells (Figure 2). This decrease was statistically significant when compared to no template control (NTC) (*p* = 0.039) and vehicle control (VC) (*p* = 0.025) treatment groups. Notably, B[a]P at higher concentrations increased *KRAS* expression (in particular at 25 µM and 10 µM) and was significantly increased from both NTC (25 µM *p* = 0.003, 10 µM *p* = 0.000) and VC groups (25 µM *p* = 0.015, 10 µM *p* = 0.005). Neither *APC* nor *CTNNB1* expression were altered by B[a]P treatment.

Phenanthrene induced some changes in *TP53, APC*, and *CTNNB1* expression when compared to the non-treated control group, however, this effect was not significant when compared to the vehicle control group. One micromolar of PHEN treatment significantly increased *KRAS* expression compared to control groups (NTC *p* = 0.005; VC *p* = 0.03).

Fluorene appeared to decrease *TP53* expression across all treatment groups, however, this relationship was not significant when compared to both control groups. Similar to B[a]P, treatments with fluorene also increased *KRAS* expression. 25 µM and 10 µM treatment groups showed increased *KRAS* expression when compared to control groups: NTC (25 µM *p* = 0.003, 10 µM *p* = 0.004) and VC (25 µM *p* = 0.016, 10 µM *p* = 0.024). FLU treatments did not alter *APC* nor *CTNNB1* expression. 

One micromolar anthracene treatment significantly increased *APC* expression when compared to both the NTC group (*p* = 0.025) and the VC group (*p* = 0.012). Furthermore, 25 µM of ANTH treatment significantly decreased *KRAS* expression compared to NTC (*p* = 0.04) and VC (*p* = 0.016). Effects of ANTH on *TP53* and *CTNNB1* were not statistically significant. 

## 3. Discussion

The current study shows the presence of potentially carcinogenic polycyclic aromatic hydrocarbons in ready-to-eat cold cut samples obtained from a local supermarket and their ability to induce genetic changes in colon cancers in vitro. Detection of PAHs in meat samples in this study using QuEChERS and GC-MS complement those of Al-Thaiban et al.’s work, which used QuEChERS (Z-sep) to prepare samples of smoked meat from Qatar and successfully analysed 16 common PAHs [25]. Our results detected 9 of 13 PAHs analysed to be present in common ready-to-eat cold cuts of meat purchased from a supermarket in Queensland, Australia. The occurrence of these PAHs suggests that contamination of pre-prepared meat products is a source of PAH exposure in the gastrointestinal tract and provides a rationale for the red/processed meat hypothesis of colorectal carcinogenesis. 

Our results showed PHEN and ANTH were present in higher levels in ham that had undergone more rounds of smoking. This is consistent with the literature that PHEN and ANTH are considered more abundant PAHs [27] and smoking of food increases PAH formation. B[a]P was detected in roast beef, roast chicken, and frankfurt samples, which corroborates well with a PAH database developed by Jakszyn et al. [28]. We hypothesise that I[1,2,3-cd] P, D[a,h]A, and B[ghi]P were absent from meat samples due to their high molecular weight and more extensive ring structure being less likely to be formed during incomplete combustion reactions. Jakszyn et al.’ s database seldom detected D[a,h]A in meat and meat products, and when present were at very low amounts [28]. Furthermore, we detected similar PAHs across ham samples, but differentiated PAHs across all samples. We hypothesise that different animal sources could yield different PAHs during the cooking process, and factors such as fat content and other ingredients used during meat preparation were not controlled for in this current study. Available literature is inconclusive regarding different meat types and PAH formation [29,30]. 

Overall, roasted turkey contained the highest total PAHs, and this may be attributed to the pattern of fat distribution throughout the meat. Roast turkey samples contained regions of pure fat, which were not observed as obviously in other meat samples. Areas of high fat concentration that are subject to high temperatures during the roasting process are more likely to create fat drippings. Fat drippings onto the source of heat undergo incomplete combustion and are a significant contributor to smoke and PAHs formation [31]. High levels of PAHs in roast turkey were in contrast to the findings in the roast chicken sample, which only contained approximately one-third of the amount of total PAHs. This is congruent with the fat-drippings rationale as the chicken meat sample did not contain any regions of pure fat whilst the turkey sample used contained a layer of fat around the meat. 

As nine different PAHs were detectable in meat samples, we chose B[a]P, PHEN, FLU, and ANTH to further investigate effects on four genes associated with colorectal cancer: *TP53*, *APC*, *CTNNB1*, and *KRAS* [24]. Varying concentrations of these PAHs were utilised as treatments for 72-h treatments as an estimate of maximal food transit time. These results showed that these PAHs that were detected in meat samples can exert genetic changes in a cell model of non-neoplastic epithelial colon cells. 

B[a]P did not show any effect on *APC* or *CTNNB1* expression in CCD841 CoN cells. Thus, its carcinogenic effects are unlikely to be via genetic changes to the APC/Wnt/β-cat pathway. However, this result is limited to one human cell line model of colon epithelial cells. There is evidence that B[a]P down-regulates *CTNNB1* and *APC* expression in the colon and stomach cells in mice [32], and the relationship between the APC/Wnt/b-cat pathway and PAHs in humans requires further investigation. We showed B[a]P was capable of increasing *KRAS* expression at higher doses (25 µM and 10 µM) and decreasing p53 mRNA expression at lower doses (1 µM). Upregulation of proto-oncogene *KRAS* is a significant finding, however, it is unlikely human ingestion of B[a]P is comparable to 25–10 µM treatment concentration. Of more clinical importance, a lower dose of B[a]P (1 µM) significantly downregulates *TP53* expression and suggests a mechanism for colorectal carcinogenesis via consumption of B[a]P-contaminated meat. Decreased p53 lowers the tumour-suppressive capability of a cell and leads to cell cycle dysregulation and uncontrolled cell growth [33]. B[a]P is capable of inducing p53 mutations in lung cancers [34] and corroborates well with our findings whereby mutated p53 is not able to be induced in response to cellular stressors. However, significant cell stress brought on by B[a]P treatment is able to transcriptionally induce p53 expression to some degree in human hepatoma cells [35] and in mouse cervical tissue [36]. Overall, B[a]P can create significant cell stress and induce *TP53* expression for cell protection. However, our results indicated that low dose B[a]P can downregulate *TP53*, perhaps by inducing a mutation in vitro, which warrants further investigation into the link between B[a]P and colorectal carcinogenesis.

Low-dose PHEN after 72 h showed significantly increased *KRAS* expression in human non-neoplastic colon cells. *KRAS* encoding for k-ras protein is a molecular on/off switch controlling cell growth and differentiation [37]. Inappropriately high expression of k-ras protein is predominately associated with an activating mutation of the *KRAS* gene, thus converting it from a proto-oncogene to an oncogene. Thus, the high level of *KRAS* expression brought on by PHEN treatment may suggest a carcinogenic link between PHEN and CRC. Similarly, moderate doses of FLU (25 µM and 10 µM) showed a similar increase in *KRAS* expression. Neither PHEN nor FLU caused other genetic changes of CRC-related genes. 

Treatment of CCD841 CoN with ANTH significantly decreased *KRAS* mRNA expression and increased *APC* expression. Although not statistically significant, ANTH also seemed to increase *TP53* and *CTNNB1* mRNA expression. Anthracene is a PAH that does not harbour a bay region and is thus theorised to be less carcinogenic compared to a bay region counterpart [38]. The bay region theory predicts that PAH-diol epoxides formed from an epoxide that is part of the bay region will have higher biological activity and thus be more mutagenic and/or tumorigenic [38]. Our results support the bay region carcinogenicity hypothesis as CCD841 CoN treated with ANTH exhibits anti-carcinogenic mRNA changes. The increased mRNA expression of *APC* after 1 µM ANTH treatment allows the cell to negatively regulate the APC/Wnt/b-cat pathway. Although low dose ANTH induced *CTNNB1*, it also slightly increased *APC*, thus it would be unlikely to lead to any overall change in activity of beta-catenin-targeted gene transcription. 

## 4. Materials and Methods

### 4.1. Chemicals

Thirteen polycyclic aromatic hydrocarbons were analysed in this study. Certified reference material EPA 525 PAH Mix B (SupelCo, Bellefonte, PA, USA) was acquired as an internal standard, containing: acenaphthylene (ACNY), anthracene (ANTH), benz[a]anthracene (B[a]A), benzo[b]fluoranthene (B[b]F), benzo[k]fluoranthene (B[k]F), benzo[ghi]perylene (B[ghi]P), benzo[a]pyrene (B[a]P), chrysene (CHRY), dibenz[a,h]anthracene (D[ah]A), fluorene (FL), indeno[1,2,3-cd]pyrene (I[123-cd]P), phenanthrene (PHEN), and pyrene (PYR). 

Sodium chloride (NaCl), magnesium sulfate (MgSO_4_), and hexane (C_6_H_14_) (HPLC grade) were obtained from Sigma-Aldrich (St. Louis, MO, USA). Hypergrade acetonitrile (ACN) was purchased from Merck (Kenilworth, NJ, USA). Sample clean-up was performed using Supel QuE Z-Sep tubes (SupelCo). Preparation of samples was performed using polypropylene tubes. Glass tubes were used during extraction to minimise interference of the samples.

### 4.2. Meat Samples

Meat samples of eight varying animal origins or preparation methods were acquired from a local supermarket delicatessen (Gold Coast, Queensland, Australia) as detailed in Table 4. Triplicates were acquired at least 2 weeks apart to ensure samples originated from different sources (knobs of ham, cuts of beef, etc.). Each sample was processed as a technical duplicate with 2 g of each sample per replicate.

### 4.3. Extraction and Clean-Up

Extraction of PAHs was performed using a type of solid phase extraction, QuEChERS (quick, easy, cheap, effective, rugged, and safe) method based on Al-Thaiban’s study [25], and summarised in Appendix A. Briefly, the homogenised samples (in 6 mL deionised water) were thoroughly mixed with 6 mL ACN and kept overnight at 4 °C. Five hundred milligrams of NaCl and 3 g MgSO_4_ were added to each sample and mixed thoroughly for 1 min. These samples were then centrifuged, and the supernatant was transferred to a QuE Z-sep tube. This was thoroughly mixed prior to further centrifugation. Once again, the supernatant was extracted, being careful to avoid the clean-up crystals in the Z-sep tubes. Samples were evaporated to dryness, reconstituted in 200 µL hexane, and stored at −20 °C.

### 4.4. GC-MS Equipment and Conditions

Gas chromatography-mass spectrometry was performed on the GCMS-TQ8030 triple quadrupole gas chromatograph (Shimadzu, Kyoto, Japan). Gas separation was performed on a 30 m Rxi-5 ms column of 0.25 mm inner diameter and 0.25µm thickness (Restek, Bellefonte, PA, USA). Helium was the carrier gas at a pressure of 92.8 kPa. A 10 µL syringe for liquid injection was used to inject the sample via split mode (ratio 5.0) at 270 °C. The initial oven temperature was 120 °C, which was increased to 270 °C over 25 min and held for 3 min. Mass spectrometric parameters were set at 230 °C as the ion source temperature, an interface temperature of 200 °C, and a solvent cut time of 2.5 min. 

### 4.5. Recovery Study

Recovery experiments were conducted for D[ah]A and B[b]F by spiking 1 µg of each into ACN at the beginning of the extraction protocol. These spiked samples were subject to the entire extraction and detection methodology across three independent experiments, each with two technical replicates. Quantification of recovery was calculated by comparing spiked and unspiked blanks.

### 4.6. Quantification of PAHs

Thirteen PAHs via EPA mix B were subjected to GC/MS as per the conditions above. Calibration curves were constructed using concentrations of 10, 25, 50, 100, and 200 ppb (x-axis) against the corresponding height (y-axis). The threshold for acceptance of a calibration curve was a correlation coefficient (R^2^) greater than 0.97. Quantification of PAH from meat sample extracts was achieved by plotting height values of samples on the calibration curve corresponding to the matched PAH analyte. 

### 4.7. Cell Culture

An in vitro model of non-neoplastic epithelial-like colon cells was created by recruiting the CCD841 CoN cell line from (ATCC CRL-1790). Cells were cultured following American Type Culture Collection (ATCC) guidelines. CCD841 CoN was cultured in EMEM (Eagle’s Minimum Essential Medium) supplemented with 10% fetal bovine serum (FBS). Cells were incubated at 37 °C with 5% CO_2_. 

### 4.8. PAH Treatment

CCD841 CoN cells were seeded in six-well plates for 24 h prior to treatment with PAHs. Benzo[a]pyrene, phenanthrene, fluorene, and anthracene were purchased from Sigma-Aldrich and prepared as 50 mM stock solutions in dimethylsulfoxide (DMSO) and stored away from the light at 4 °C. PAH stock solutions (50 mM) were transferred to a dosing plate with final concentrations of 50 µM, 25 µM, 10 µM, and 1 µM diluted in EMEM + 10% fetal bovine serum (FBS) media and used to treat six-well plates. Non-treated control (NTC) treatments involved EMEM + 10% FBS media in place of PAH treatment, and vehicle control (VC) treatment consisted of treating cells with 0.2% DMSO in place of PAH. Cells were incubated for 72 h prior to downstream experiments. 

### 4.9. RNA Extraction and Quantitative Real-Time Polymerase Chain Reaction (rtPCR)

CCD841 CoN^+PAH^ cells were harvested via trypsinisation and RNA was extracted using miRNeasy mini kit (Qiagen, Valencia, CA, USA) following the manufacturer’s protocol. Total RNA purity and quantification was performed using Nanodrop Spectrophotometer, measured via 260/280 and ng/µL, respectively. This RNA was converted to cDNA using SensiFAST cDNA synthesis kit (Meridian Bioscience, Cincinnati, OH, USA) following the manufacturer’s guidelines. cDNA was again measured using a Nanodrop spectrophotometer (BioLab, Milford, MA, USA) and diluted to a 100 ng/µL working concentration. The mRNA expressions of *TP53*, *APC*, *CTNNB1*, and *KRAS* were investigated using rtPCR (QuantStudio, Thermostat Fisher Scientific, Waltham, MA, USA). *Glyceraldehyde-3-phosphate dehydrogenase (GAPDH)* was used as a housekeeping gene. Primer sequences are detailed in Appendix A.

### 4.10. Statistical Analysis

Analyses with PAH concentrations in meat samples were performed in GraphPad Prism 9.0 (San Diego, CA, USA) using unpaired *t*-tests. rtPCR analysis was performed as previously reported [39]. Statistical analysis of gene expression study was performed by comparing the difference in C_t_ values but presented graphically as fold changes. Statistical significance level was considered at *p* < 0.05.

## 5. Conclusions

This study demonstrated a link between PAH compounds in processed meat and alteration of colorectal cancer genes. Our study validated the use of QuEChERS method for PAH sample preparation from meat sources, which allows for quick, easy, and cheap extraction and clean-up for future studies. Using this method, we showed that commonly consumed ready-to-eat meat samples, which did not undergo any further processing or cooking, contain various polycyclic aromatic hydrocarbons. Congruent with the available literature, phenanthrene, anthracene, and fluorene were most detected among meat samples. Furthermore, benzo[a]pyrene was present in three of the eight meat samples investigated. The four PAHs showed capabilities of colorectal cancer-related gene alterations in a non-neoplastic human colon cell line. This study may provide direction for future investigations regarding processing of meat and the mechanisms by which red and processed meat intake might confer colorectal cancer risk.

## Figures and Tables

**Figure 1 ijms-22-10959-f001:**
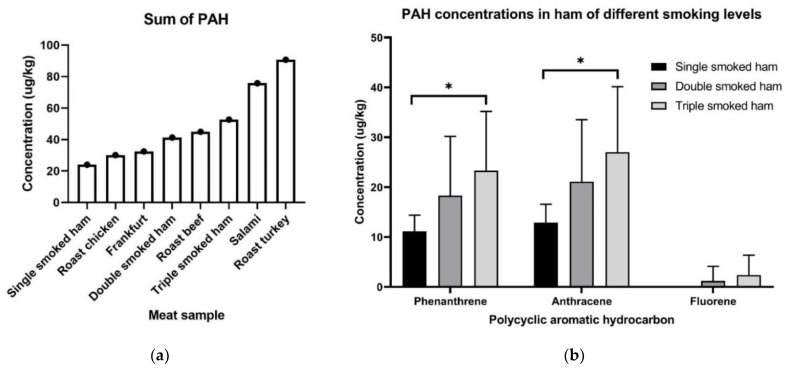
Graph (**a**) shows the total PAH levels detected in each meat sample of single-smoked ham containing the least PAH content and roast turkey having the highest total PAH while; (**b**) specifically shows that triple-smoked ham has significantly higher concentrations of phenanthrene and anthracene than single-smoked ham. * = *p* < 0.05.

**Figure 2 ijms-22-10959-f002:**
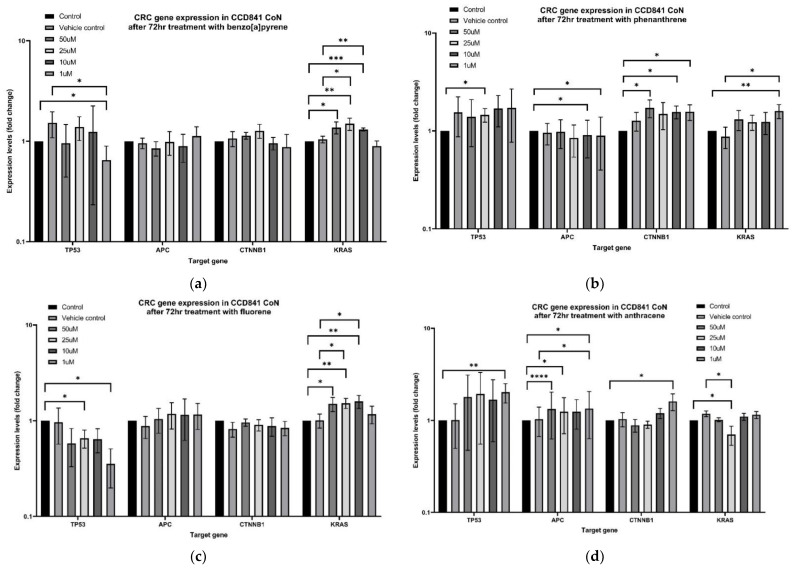
mRNA expression changes of *TP53*, *APC*, *CTNNB1*, and *KRAS* in CCD841 CoN cell line after 72-h treatment with (**a**) B[a]P; (**b**) PHEN; (**c**) FLU; and (**d**) ANTH. Statistical significance was considered at *p* > 0.05. * = *p* < 0.05; ** = *p* < 0.01; *** = *p* < 0.001; **** = *p* < 0.0001.

**Table 1 ijms-22-10959-t001:** IARC groups of polycyclic aromatic hydrocarbons are investigated in this study. Group 1: carcinogenic; group 2A: probably carcinogenic; group 2B: possibly carcinogenic; group 3: not classifiable; N/A: no information available.

Polycyclic Aromatic Hydrocarbon	IARC Group
Acenaphthylene	N/A
Anthracene	3
Benz[a]anthracene	2B
Benzo[a]pyrene	1
Benzo[b]fluoranthene	2B
Benzo[ghi]perylene	3
Benzo[k]fluoranthene	2B
Chrysene	2B
Dibenz[a,h]anthracene	2A
Fluorene	3
Indeno[1,2,3-cd]pyrene	2B
Phenanthrene	3
Pyrene	3

**Table 2 ijms-22-10959-t002:** Equation characteristics for PAH quantification. Curve equations were determined using five points of identification (10, 25, 50, 100, and 200 ppb). All curve equations had an *r*-value greater than 0.993.

Standards	Mass (g)	Concentration Range (ppb)	Curve Equation	Regression Coefficient
Acenaphthylene	152.00 > 150.10	10–200	$f\left(x\right)=0.2210x^{2}+45.5552x+0.0000$	0.9977
Fluorene	165.00 > 163.10	10–200	$f\left(x\right)=0.3354x^{2}+49.7426x+0.0000$	0.9971
Phenanthrene	178.00 > 152.10	10–200	$f\left(x\right)=0.4970x^{2}+89.7263x+0.0000$	0.9948
Anthracene	178.00 > 176.10	10–200	$f\left(x\right)=0.4223x^{2}+74.9640x+0.0000$	0.9967
Pyrene	202.00 > 200.10	10–200	$f\left(x\right)=0.3417x^{2}+50.8238x+0.0000$	0.9933
Benzo[a]anthracene	228.00 > 226.10	10–200	$f\left(x\right)=0.4917x^{2}+83.1846x+0.0000$	0.9977
Chrysene	228.00 > 226.00	10–200	$f\left(x\right)=0.6500x^{2}+85.4511x+0.0000$	0.9960
Benzo[b]fluoranthene	252.00 > 250.00	10–200	$f\left(x\right)=0.2843x^{2}+35.2259x+0.0000$	0.9961
Benzo[k]fluoranthene	252.00 > 250.10	10–200	$f\left(x\right)=0.4565x^{2}+61.2200x+0.0000$	0.9963
Benzo[a]pyrene	252.00 > 250.10	10–200	$f\left(x\right)=0.3910x^{2}+39.8408x+0.0000$	0.9945
Indeno[1,2,3-cd]pyrene	276.00 > 274.00	10–200	$f\left(x\right)=0.2577x^{2}+14.8013x+0.0000$	0.9932
Dibenz[a,h]anthracene	278.00 > 276.10	10–200	$f\left(x\right)=0.4289x^{2}+30.2481x+0.0000$	0.9961
Benzo[ghi]perylene	276.00 > 274.00	10–200	$f\left(x\right)=0.2677x^{2}+23.2594x+0.0000$	0.9951

**Table 3 ijms-22-10959-t003:** Recovery of PAHs as determined comparing spiked blanks and unspiked blanks.

PAH	Spiked Concentration ppb	Mean Concentration Detected ppb	Unspiked Blank Concentration ppb	Mean Recovery %
Benzo[b]fluoranthene	1000	943.5	0	94.4
Dibenz[a,h]anthracene	1000	744.5	0	74.5

**Table 4 ijms-22-10959-t004:** Details of meat samples obtained from a local supermarket, including procedures of meat cooking/preparation.

Sample Name	Meat	Preparation Method	Number of Replicates (Technical; Biological)
Roast turkey	Turkey	Roasted	2; 3
Roast beef	Beef	Roasted	2; 3
Roast chicken	Chicken	Roasted	2; 3
Frankfurt	Pork	Smoked meat trimmings	2; 3
Single smoked ham	Pork	Single smoked	2; 3
Double smoked ham	Pork	Double smoked	2; 3
Triple smoked ham	Pork	Triple smoked	2; 3
Salami	Pork	Fermented and matured with smoke	2; 3

## Data Availability

The data presented in this study are available in Polycyclic Aromatic Hydrocarbons Detected in Processed Meats Cause Genetic Changes in Colorectal Cancers and its Appendix A within.

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
