# Peer review of "Polycyclic Aromatic Hydrocarbons Detected in Processed Meats Cause Genetic Changes in Colorectal Cancers"

_ijms, 2021, doi:10.3390/ijms222010959_

Round 1
Reviewer 1 Report
The manuscript needs a reorganization:
- The section describing the methodology or materials and methods must be reported after the introduction.
- The state of the art reported in the introduction must be extended, currently the introduction describes a part of the experimental part.
- The general objective of the study must be defined both in the abstract and in the introduction
How do the authors justify the reasons for the presence of differentiated PAHs in different foods based on the evidence of their studies and existing literature?
Author Response
- Comment 1: The section describing the methodology or materials and methods must be reported after the introduction.
Response: Thank you for this suggestion. We agree that Materials and Methods would usually be reported after the introduction, however the IJMS guidelines for a research manuscript reports the Materials and Methods section after the Results and Discussion sections. As such, we have found it difficult to address this comment, and have decided to do our best to follow the journal guidelines, thus we have kept the Materials and Methods section in its current place. - Comment 2: The state of the art reported in the introduction must be extended, currently the introduction describes a part of the experimental part.
Response: Thank you for noticing this. We agree that we could have more adequately reported on the current state of the field and have extended on this within the introduction (lines 59-66). - Comment 3: The general objective of the study must be defined both in the abstract and in the introduction.
Response: We agree that the objective of the study should be stated in both the abstract and introduction. Within the abstract, we have included a statement defining the aims on lines 12-14, and a similar statement in the introduction on page 2 (lines 71-74).
- Comment 4: How do the authors justify the reasons for the presence of differentiated PAHs in different foods based on the evidence of their studies and existing literature?
Response: This is a great comment, thank you for asking this question. The existing literature is overall inconclusive regarding differentiated PAHs in different meats. Our results suggest that different meat types (i.e., chicken, beef, pork) yield different PAHs, however this could be from a myriad of reasons such as how the meat was prepared, the amount of fat present, other ingredients etc that were not controlled for in our study. To address this, we have added a paragraph on page 6 (lines 178-188). Thank you again, we appreciate such questions and the depth of thought you have put into our study.
Reviewer 2 Report
This manuscript presents the results of laboratory experiments on the roles of various PAHs found in meat on expression of key CRC-related genes. The experiments were done carefully and the results are very interesting.
I suggest searching scholar.google.com with the search string “Polycyclic aromatic hydrocarbons, meat, colorectal cancer” and same with “genetic changes” to find some of the historical publications on this topic. The number of citations to each publication will be shown, giving an indication of the importance.
Meat consumption is a risk factor for many types of cancer.
Meat consumption and cancer risk: a case-control study in Uruguay.
Aune D, De Stefani E, Ronco A, Boffetta P, Deneo-Pellegrini H, Acosta G, Mendilaharsu M.Asian Pac J Cancer Prev. 2009 Jul-Sep;10(3):429-36.
Perhaps the role of PAHs from meat (and atmospheric pollutants) for other cancers could also be discussed briefly (just a suggestion).
Polycyclic aromatic hydrocarbons and digestive tract cancers: a perspective
DL Diggs, AC Huderson, KL Harris… - … science and health …, 2011 - Taylor & Francis
… The formation and occurrence of polynuclear aromatic hydrocarbons associated with food … 1993.
Association of PAH-DNA adducts in peripheral white blood cells with dietary exposure to
polyaromatic hydrocarbons … “Carcinogenic polycyclic aromatic hydrocarbons in food” …
Cited by 186 Related articles All 8 versions
A review of airborne polycyclic aromatic hydrocarbons (PAHs) and their human health effects
KH Kim, SA Jahan, E Kabir, RJC Brown - Environment international, 2013 - Elsevier
… of airborne and dermal exposure to polycyclic aromatic compounds in asphalt … White, 2002 PA
WhiteThe genotoxicity of priority polycyclic aromatic hydrocarbons in complex … Organization), 2003
WHO (World Health Organization)Polynuclear aromatic hydrocarbons in drinking …
Cited by 1476 Related articles All 11 versions
The relationship between genetic damage from polycyclic aromatic hydrocarbons in breast tissue and breast cancer
A Rundle, D Tang, H Hibshoosh, A Estabrook… - …, 2000 - academic.oup.com
… 11 International Agency for Research on Cancer (1983) Polynuclear aromatic compounds, part
1 … WA and Gould,MN ( 1986. ) Interspecies comparison of polycyclic aromatic hydrocarbon
metabolism in … ( 1990. ) Genotoxic effects of five polycyclic aromatic hydrocarbons in human …
Cited by 241 Related articles All 7 versions
Polycyclic aromatic hydrocarbons and digestive tract cancers: a perspective
DL Diggs, AC Huderson, KL Harris… - … science and health …, 2011 - Taylor & Francis
… The formation and occurrence of polynuclear aromatic hydrocarbons associated with food … in
peripheral white blood cells with dietary exposure to polyaromatic hydrocarbons … lung cytochromes
P450 1A1, 1A2, and 1B1 by polycyclic aromatic hydrocarbons and polychlorinated …
Cited by 186 Related articles All 8 versions
Polycyclic aromatic hydrocarbons in the diet
DH Phillips - Mutation research/genetic toxicology and …, 1999 - Elsevier
… G. Grimmer, J. Jacob, AE Johnston Changes in the polynuclear aromatic hydrocarbon content
of … white blood cells with dietary exposure to polyaromatic hydrocarbons. Environ … Adsorption of
polycyclic aromatic hydrocarbons (PAHs) by cellulosic aerogels during smoked pork …
Cited by 1096 Related articles All 9 versions
Author Response
- Comment 1: This manuscript presents the results of laboratory experiments on the roles of various PAHs found in meat on expression of key CRC-related genes. The experiments were done carefully, and the results are very interesting.
Response: Thank you very much, this is very kind to say! - Comment 2: I suggest searching scholar.google.com with the search string “Polycyclic aromatic hydrocarbons, meat, colorectal cancer” and same with “genetic changes” to find some of the historical publications on this topic. The number of citations to each publication will be shown, giving an indication of the importance, i.e. meat consumption is a risk factor for many types of cancer. Response Thank you, we think this is a great idea and we have incorporated some of our searches based on your suggestion in lines 59-67 of the introduction.
- Comment 3: Perhaps the role of PAHs from meat (and atmospheric pollutants) for other cancers could also be discussed briefly (just a suggestion).
Response: Thank you for this suggestion, we also believe it is important to link PAHs to cancers in general. To do this, we have included a brief sentence in the introduction (lines 62-64) that links non-dietary environmental PAHs to lung and breast cancers.
Round 2
Reviewer 1 Report
I agree to maintain the organization of the manuscript given that the format of the IJMS is as given by the authors.
All the other improvements have been made; therefore the paper can be accepted in the present form.